# Endovascular Treatment Outcomes for TASC C and D Lesions in Chronic Peripheral Arterial Disease: A Retrospective Study and Literature Review

**DOI:** 10.3390/biomedicines13112771

**Published:** 2025-11-13

**Authors:** Manfredi Agostino La Marca, Salvatore Bruno, Giovanni Gagliardo, Ettore Dinoto, Rosa Federico, Felice Pecoraro, Domenico Mirabella

**Affiliations:** 1Vascular Surgery Unit, AOUP Policlinico “P. Giaccone”, 90127 Palermo, Italy; manfredi.a.lamarca@gmail.com (M.A.L.M.); salvatorebruno93@gmail.com (S.B.); gagliardo.gnn@gmail.com (G.G.); rosa6federico@gmail.com (R.F.); felice.pecoraro@unipa.it (F.P.); dmirabella@live.it (D.M.); 2Department of Surgical, Oncological and Oral Sciences, University of Palermo, 90127 Palermo, Italy

**Keywords:** peripheral artery disease, TASC, critical limb ischemia, peripheral revascularization

## Abstract

**Background**: Peripheral Artery Disease (PAD) of the lower extremities is a prevalent manifestation of atherosclerotic disease, significantly affecting individuals aged 55–70, with a global incidence of 4–12%. Major risk factors include smoking, diabetes mellitus, hypertension, dyslipidemia, and chronic kidney disease, all contributing to endothelial damage and subsequent plaque progression. This retrospective study examines the outcomes of endovascular treatment for TASC C and D lesions, which are complex cases that have historically required surgical intervention. **Methods**: From June 2022 to September 2023, 48 patients were analyzed, with a mean age of 67.48 years; 37.5% were female. Statins were administered to 64.6% of patients, and 93.8% received antiplatelet therapy. Endovascular procedures included balloon angioplasty, stenting, and the use of drug-eluting balloons (DEB), employing varying access routes, primarily via percutaneous approaches. **Results**: The study revealed a 12-month primary patency rate of 75.8% and a secondary patency rate of 95.5%, highlighting the effectiveness of follow-up interventions. Complications occurred in 10.4% of cases, with a perioperative mortality rate of 0%. Notably, 29.2% of patients required amputation, reflecting the severity of PAD. **Conclusions**: The outcomes demonstrate that endovascular treatment may be a viable alternative for managing TASC C and D lesions, offering satisfactory clinical outcomes and an acceptable safety profile. Continuous monitoring and interdisciplinary evaluations are essential for optimizing patient care and minimizing complications. As endovascular technologies advance, their role in treating severe peripheral arterial disease is likely to expand.

## 1. Introduction

The manifestations of atherosclerotic disease affecting the peripheral arterial system occur in approximately 4–12% of individuals aged 55–70 years [1]. The major risk factors for developing Peripheral Artery Disease (PAD) include cigarette smoking and diabetes mellitus, both of which can increase the risk of progression to critical ischemia by two to four times compared to the general population [2]. In these patients, reduced blood flow to the extremities, coupled with microcirculatory damage, can result in major amputation, with an estimated incidence rate of 30.7% at 12 months [3]. When medical therapy fails to control the progression of PAD and critical ischemia manifests, thereby increasing the risk of amputation, vascular surgery is often employed through open, endovascular, or hybrid revascularization procedures in an attempt to bring the disease to a less clinically significant stage [4]. Advances in endovascular techniques have significantly decreased the invasiveness of revascularization procedures. Consequently, there has been a concerted effort to improve patient outcomes by classifying vascular lesions more effectively [5,6,7]. The TASC (TransAtlantic Inter-Society Consensus) classification, established by the 2007 Inter-Society Working Group Consensus on the management of peripheral arterial disease (TASC II), guides treatment decisions. It recommends endovascular treatments for the simpler lesions (TASC A and B), while surgical intervention is indicated for longer, more complex lesions (TASC C and D).

This retrospective study presents the outcomes of endovascular treatment for TASC C and D lesions, complex cases traditionally managed with open surgery that, thanks to technological advancements, can now potentially be treated with less invasive approaches.

## 2. Materials and Methods

The study was conducted at the Vascular Surgery Unit of the P. Giaccone University Hospital in Palermo, involving patients treated between June 2022 and September 2023. The patients included in the study were affected by peripheral arterial disease (PAD), experiencing severely limited walking distances of just a few meters or presenting with critical limb ischemia. This is a non-randomized retrospective study aimed at describing our experience using endovascular treatment for TASC C and D lesions (Figure 1 and Figure 2), for which surgical intervention is typically indicated. All patients were collected and inserted into standardized piloted forms. This study was performed in agreement with the Declaration of Helsinki, and the STROBE guidelines for reporting observational studies were followed [8].

Patient demographics, including sex and age, as well as key risk factors and comorbidities such as hypertension, smoking, diabetes mellitus, dyslipidemia, chronic renal insufficiency, the need for dialysis, and any relevant history of cerebrovascular disease, were recorded. The findings of this study are based on previous research in the fields of vascular disease, ischemic heart disease with prior PCI (percutaneous coronary intervention)/CABG (coronary artery bypass graft) interventions, and tumors. Additionally, we assessed the pharmacological therapies administered to patients and the history of any prior surgical or endovascular procedures.

Patients were classified according to the Rutherford classification system and the TASC classification based on the type of atherosclerotic lesions. They were also categorized according to the American Society of Anesthesiology (ASA) classification, and the type of anesthesia used for each intervention was documented [9]. Only patients classified as TASC C and D and treated via the endovascular approach were included in this study; those classified as TASC A and B, as well as patients who underwent surgical or hybrid treatments, were excluded. The preoperative assessment included a duplex ultrasound (DUS). A computed tomography angiography (CTA) was utilized for all cases involving aortic-iliac disease.

For each procedure, we evaluated the duration of the intervention, the need for blood transfusions, the length of hospital stay, and any admissions to intensive care. Follow-up assessments were conducted through clinical examination and DUS evaluations at one week post-treatment and 1 month, after 6 months, and every 12 months thereafter, during which we monitored for the onset of early, acute, sub-acute, and late complications, as well as the need for reoperation and subsequent amputations of the lower limbs. The primary aim of this study is to assess the efficacy and outcomes of endovascular treatment for TASC C and D lesions. This study did not require approval from the hospital’s ethics committee, and all patients provided informed consent before treatment. Before undergoing surgery, each patient underwent preoperative laboratory tests (with particular attention to hemoglobin levels and renal function), a preoperative cardiology assessment, a chest X-ray, and an anesthesiological evaluation.

For statistical analysis, means and SDs or medians and ranges were reported for continuous non-parametric data; absolute values and percentages were reported for categorical data. Differences in preoperative and postoperative outcomes were assessed using the Student’s *t*-test. A bivariate test was used to assess the relationship significance for correlation analysis. These values were log-transformed for discrete skewness. We tested for linearity using a test for linear trends across the quartiles. Statistical analysis was performed using SPSS 16.0 (IBM, New York, NY, USA).

## 3. Results

The study comprised 48 patients, of whom 18 were female (37.5%), with a mean age of 67.48 years (SD 12, IQR 56–76). The median follow-up was 12 (mean: 12; r: 12–26) months.

About the location of the lesion, two groups have been identified: patients with lesions in the aorto-iliac region and lesions in the femoropopliteal region. Baseline demographic and clinical features are reported in Table 1.

Among the participants, 31 patients (64.6%) were receiving statin therapy, and 45 patients (93.8%) were on antiplatelet medication. Of these, 22 patients (45.8%) were on dual antiplatelet therapy (DAPT) with acetylsalicylic acid (ASA) and clopidogrel, while 23 patients (47.9%) were on single antiplatelet therapy (SAPT), 15 patients (31.3%) on ASA and 8 patients (16.7%) on clopidogrel. Thromboembolic prevention therapy (TAO), characterized by a single administration of 4.000 IU of heparin, was given to 6 patients (12.5%) with significant mobility difficulties; among them, 2 were on DAPT and 2 on SAPT. Additionally, 7 patients (14.5%) received non-vitamin K antagonist oral anticoagulants (NOACs), with 2 patients on DAPT and 4 on SAPT (Table 2).

Previous vascular surgery for atherosclerotic disease of the lower limbs was noted in 25 patients (52.1%), with 18 (37.5%) undergoing endovascular procedures and 7 (14.6%) surgical interventions. The interventions previously carried out on these patients differed from those for which the subjects were included in the study. The preoperative evaluation for all patients included Doppler ultrasound (ECD), while CT angiography was performed in 21 patients (43.8%). Intraoperative angiography was conducted to assess arterial lesions and confirm the planned treatment strategy. An anatomical assessment of lesion localization and classification according to the TASC criteria was based on the integration of preoperative diagnostic data (DUS, CTA, and angiography). A total of 23 patients (47.9%) were classified as TASC class C, and 25 patients (52.1%) as TASC class D. All subjects were diagnosed with PAD, most commonly with multiple lesions; based on the surgical risk according to the ASA classification, 47 patients were classified as ASA III (97.9%), and 1 patient was ASA IV (2.1%). The initial clinical presentation was categorized as Rutherford 3, 4, 5, and 6 in 7 (14.6%), 10 (20.8%), 26 (54.2%), and 5 (12.5%) patients, respectively (Table 3).

A total of 210 arterial lesions were examined, distributed as shown in Table 4.

In 44 cases, locoregional anesthesia was employed (91.7%), while general anesthesia was used in 3 cases (6.2%) and a nerve plexus block in 1 case (2.1%). Seven different types of endovascular interventions were performed, each selected based on the patient’s clinical condition and the characteristics of the arterial lesions. In 32 cases (66.7%), balloon catheter percutaneous transluminal angioplasty (PTA) was performed, with 12 cases as isolated procedures (Figure 3 and Figure 4) and 20 cases of Hybrid procedures (Figure 5). A total of 24 patients (50%) received stents (Figure 6). Medicated balloons were utilized in 24 patients (50%), with 3 out of 4 procedures performed in conjunction with other techniques. In 6 cases (12.5%), intravascular lithotripsy was performed using shockwave technology (Figure 7), 4 cases (8.3%) involved Debulking using a rotational atherotome (Figure 8), and only 2 cases (4.1%) required implantation of an Unibody aortic endoprosthesis (AFX endovascular AAA system; Endologix Inc., Irvine, CA, USA) (Figure 9). Notably, the data indicate a frequent combination of different techniques, with 30 cases (62.5%) employing multiple approaches (Table 5).

Concerning access routes for endovascular techniques, 6 patients (12.5%) underwent a surgical femoral approach, while 42 patients (87.5%) received percutaneous access 33 (68.8%) via ipsilateral femoral access, 8 (16.7%) via contralateral femoral access, and 1 (2.1%) using retrograde access from the PTA employing the Safari technique. The average duration of the interventions was 94.69 min.

Various materials were used during the endovascular procedures. A total of 41 stents were implanted, with an average length of 61.6 mm and an average diameter of 6.9 mm. Covered stents were primarily used to treat lesions at the aortic bifurcation and common iliac arteries, with balloon-expandable stents being the predominant choice in these locations. For the external iliac arteries, bare metal stents, mostly self-expanding, were favored. In the femoral region, self-expanding bare metal stents were commonly deployed, while covered self-expanding stents were reserved for cases involving the popliteal artery. For PTA, 138 balloon catheters were utilized, including 89 (64.5%) standard catheters with an average length of 109.3 mm and an average diameter of 4.6 mm, 42 (30.4%) drug-eluting balloons (DEB) with an average length of 131.5 mm and an average diameter of 4.1 mm, and 7 (5.1%) shockwave balloons. Rotational atherectomy was employed in four patients, while Prostyle-type percutaneous closure systems were used in 5 cases, and a 6 mm Spyder filter was implanted in 1 case.

The perioperative mortality rate at 30 days was 0%. Surgical complications were categorized based on their timing. Early complications (occurring within 24 h post-surgery) were observed in 2 patients (4.2%), including 1 case of aortic dissection and 1 case of a pseudo-aneurysm at the site of percutaneous access. Acute complications (within 30 days post-surgery) affected 6 patients (12.5%), with 2 cases of stent occlusion, 2 cases of surgical wound dehiscence, 1 case of early restenosis of the SFA, and 1 case of Acute Renal Failure. Sub-acute complications (1–12 months post-surgery) were reported in 2 patients (4.2%), with one case of stent occlusion and one case of leg vessel restenosis. Late complications involved 1 patient (2.1%) with worsening ischemic and trophic lesions in the foot.

The primary patency rate at 12 months was 75.8%, with a slight advantage observed in the femoropopliteal region, which has demonstrated lasting effectiveness over time. In contrast, interventions performed in the aorto-iliac region showed decreased effectiveness after the second year, primarily due to a high rate of intrastent restenosis. The secondary patency rate was 95.5% after 1 year, with a reduction to 79.6% at two years due to the need for a higher reintervention rate of the aorto-iliac district (Figure 10 and Figure 11). A survival rate of 88.2% in the absence of significant changes over time (Figure 12). The reintervention rate was 16.7%, involving 8 patients.

In 14 cases (29.2%), amputation was deemed necessary: 9 patients (18.7%) underwent minor amputations (amputation at the level of the foot), and 5 patients (10.4%) underwent major amputations (below or above the knee) (Figure 13 and Figure 14).

## 4. Discussion

The management of complex arterial lesions classified as TASC C and D poses a significant challenge in clinical practice due to their anatomical complexity and the severity of PAD. Our study aimed to evaluate the outcomes of endovascular treatment for these lesions, focusing on clinical results and short- and long-term success rates. The collected data showed that endovascular treatment of TASC C and D lesions is effective, though success varies depending on factors such as the degree of calcification in the treated segment and patient adherence to medication, follow-up visits, and risk factor management such as diabetes and smoking. Our findings indicated a primary patency rate of 75.8% at 12 months. We observed a primary patency rate of 75.8% at 12 months—lower than rates reported for less complex lesions (TASC A and B) but still significant given the lesion complexity (Figure 15). Notably, only 50% of patients received stent implantation despite the presence of TASC C and D lesions. Calcification and plaque extent pose key challenges by reducing the efficacy of DEB and stents, increasing restenosis risk and need for reintervention. Meticulous vessel preparation is critical to optimize outcomes and improve primary patency. Increased use of vessel preparation devices before PTA has enabled a broader DEB deployment, reducing stent implantation rates. The highest stenting rate (94.1%) occurred in the aortoiliac region, which also had a higher reintervention rate mainly due to intrastent restenosis. Clinical evidence supports the use of covered stents at the aortic bifurcation and common iliac arteries, where they prevent plaque prolapse and embolization, especially important in large vessels with complex anatomy [10]. Balloon-expandable covered stents are preferred here for precise deployment and radial strength, while self-expanding bare metal stents are favored in the external iliac and femoral arteries due to their flexibility and ability to accommodate vessel movement, reducing fracture risk and improving patency [11].

Two cases involved the AFX unibody stent-graft for aortic bifurcation reconstruction. Data from the MURUSSIAS Registry support the AFX system as a durable and safe solution for complex aortoiliac occlusive disease, offering superior anatomical conformity, stable fixation, and low complication rates such as endoleaks, even in acute cases. This enhances both technical success and long-term vessel patency [12,13]. Secondary patency at 12 months was 95.5%, indicating that additional interventions can significantly improve long-term outcomes. The amputation rate was 29.2%, mostly minor amputations, which compares favorably with the 30.7% major amputation rate reported in the literature for critical ischemia. Notably, 87.5% of our patients had critical ischemia (Rutherford > 4) with 66.7% presenting trophic lesions at admission, and most amputations occurred within the first year. Our findings align with literature showing increased endovascular treatment for PAD and decreased open surgery without significant changes in amputation rates. Studies by Nasr et al. and Dosluoglu et al. demonstrate comparable patency rates among endovascular, open, and hybrid approaches, with endovascular methods offering lower perioperative risks and reduced hospital stays despite higher material costs. The BASIL trial confirmed similar survival without amputation between bypass and angioplasty but noted higher hospitalization costs for surgery [14,15,16,17]. Minor amputations observed in this study resulted from successful revascularization that averted the need for more extensive tissue removal. In cases requiring major amputation, patients presented with severely compromised baseline conditions that either prevented optimal revascularization or led to a gradual loss of the procedure’s benefits over time.

Furthermore, advancements in modern angiographic technology have significantly reduced the radiation exposure for both operators and patients, as well as the volumes of contrast material required for optimal image acquisition, thereby mitigating associated complications [18,19].

Ahn et al. reported a 93.3% technical success and 94.9% two-year patency for endovascular treatment of aortoiliac TASC C and D lesions, though they discouraged brachial access due to technical difficulties. Our own experience with 167 patients supports endovascular and hybrid approaches, showing one-year freedom from occlusion at 70%, secondary patency at 80%, major amputation at 2.4%, and minor amputation at 41.9%. Restenosis was higher in aortoiliac lesions, likely due to less vessel preparation and greater stent use compared to the femoropopliteal region [7,20]. Kaplan–Meier analysis indicates high limb salvage rates after revascularization, with major amputation reduction and increased minor amputations, particularly in femoropopliteal patients with diabetes-related microcirculatory issues.

The outcomes reveal that endovascular treatment for TASC C and D lesions is a valid therapeutic approach, yielding satisfactory clinical results and an acceptable safety profile. Despite the inherent complexity of these lesions, the adoption of advanced endovascular techniques has resulted in high technical success rates and a significant reduction in peri-vascular complications.

However, the TASC classification does not fully reflect advances in surgical and endovascular methods, which now allow minimally invasive treatment of lesions once addressed only by open surgery [21,22,23,24]. Additionally, Faglia et al. have noted that in clinical practice, patients classified as TASC C and D are often prioritized for endovascular options, with open treatments reserved for cases of failure or when the patient’s overall condition is more suitable for invasive procedures [25,26]. Moreover, it is important to recognize that integrating both open and endovascular approaches can further improve patient outcomes. Graziani et al. highlight that multilevel involvement is quite prevalent among diabetic patients with ischemic foot lesions. They point out that a significant limitation of the TASC classification is its failure to account for this common anatomical situation, as it evaluates each lesion in isolation [27]. Consequently, the TASC categories do not adequately reflect the severity of the disease or guide appropriate treatment strategies, especially in the subgenicular district, where endovascular approach is a first-line therapeutic choice [28,29]. Although TASC has faced criticism from numerous authors, many continue to utilize it as a standard framework for classifying disease severity in a common international language [30].

Complications occurred in over 10% of cases, mainly acute thrombosis, vascular perforation, and surgical conversions, consistent with literature. Careful patient selection and planning are essential. The GLASS classification offers improved predictive value by considering calcium distribution within arterial segments, aiding tailored revascularization strategies [31,32,33].

In summary, evidence supports the effectiveness of endovascular treatment for complex arterial lesions, emphasizing the need for meticulous patient selection and procedural planning to optimize outcomes and reduce risks.

## 5. Conclusions

The results of this study confirm that endovascular treatment represents a valid and safe option for patients with TASC C and D lesions, particularly for those at high surgical risk. However, to optimize clinical outcomes, a multidisciplinary approach and rigorous, continuous follow-up are essential. Technological advancements in endovascular interventions, including new support devices and innovative stents, promise to further improve mid- and long-term results. Continuous monitoring is crucial to promptly prevent and manage potential recurrences or late complications, thereby ensuring an improved quality of life for patients. Our experience highlights that, with increasing clinical expertise and evolving technology, endovascular techniques are assuming an increasingly central role in the management of complex peripheral arterial lesions. It is therefore desirable that future guidelines explicitly recognize the importance and effectiveness of these techniques in treating highly complex peripheral arterial disease.

## Figures and Tables

**Figure 1 biomedicines-13-02771-f001:**
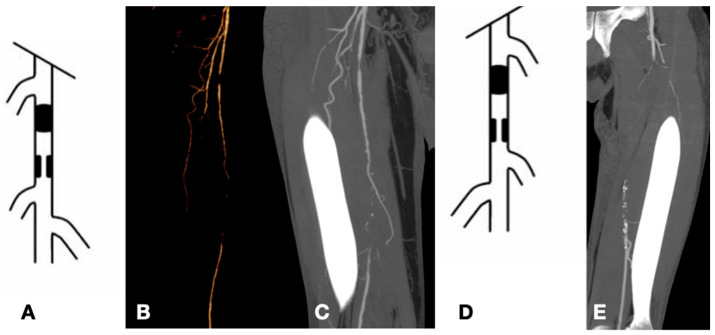
Examples of TASC C and D lesions in the femoropopliteal district: TASC C (**A**–**C**) with occlusions of the superficial femoral artery less than 20 cm in length; TASC D (**D**,**E**) with occlusions greater than 20 cm.

**Figure 2 biomedicines-13-02771-f002:**
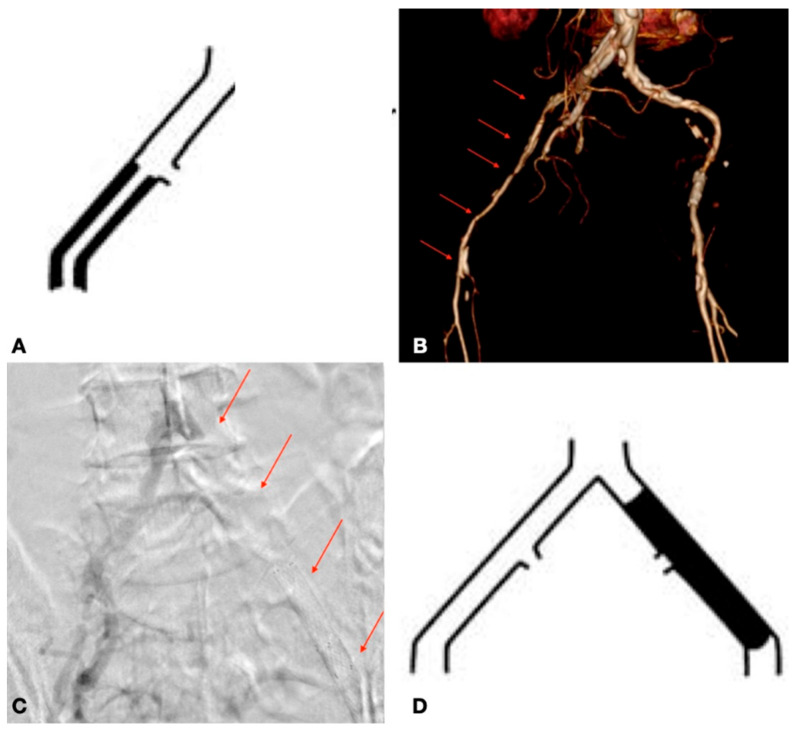
Examples of TASC C and D lesions in the aortoiliac district: TASC C (**A**,**B**) characterized by multiple stenoses of the external iliac artery extending to the common femoral artery; TASC D (**C**,**D**) involving occlusion of both the common iliac artery and the external iliac artery.

**Figure 3 biomedicines-13-02771-f003:**
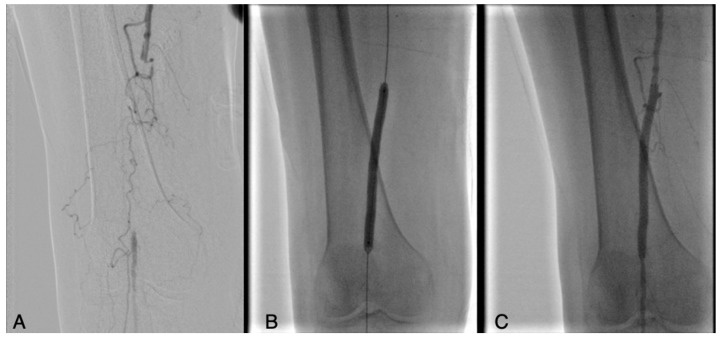
Occlusion of the femoropopliteal junction (**A**) treated with PTA (**B**) and complete ricanalization (**C**).

**Figure 4 biomedicines-13-02771-f004:**
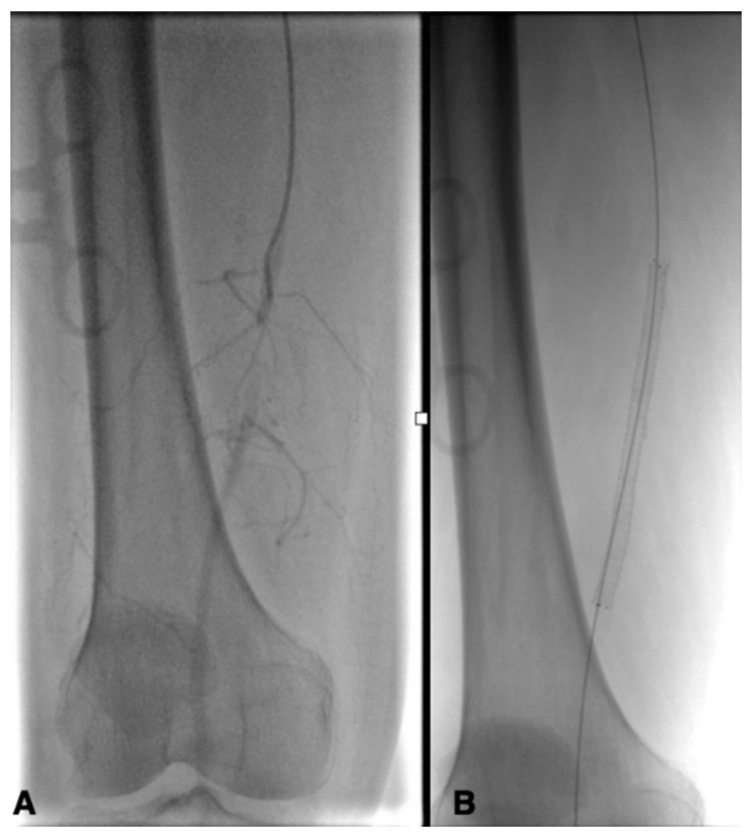
Occlusion of the distal SFA (**A**) treated with stenting (**B**).

**Figure 5 biomedicines-13-02771-f005:**
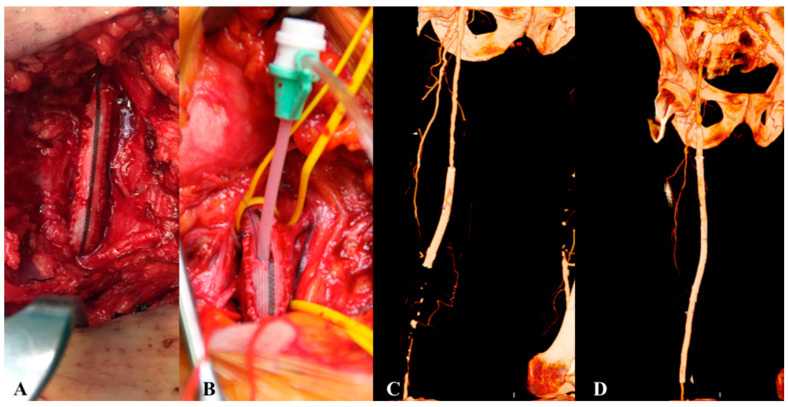
Hybrid procedure of femoral endarterectomy with patch angioplastym (**A**,**B**) and revascularization of the superficial femoral artery ((**C**) before PTA con occlusion intra stent, (**D**) after PTA).

**Figure 6 biomedicines-13-02771-f006:**
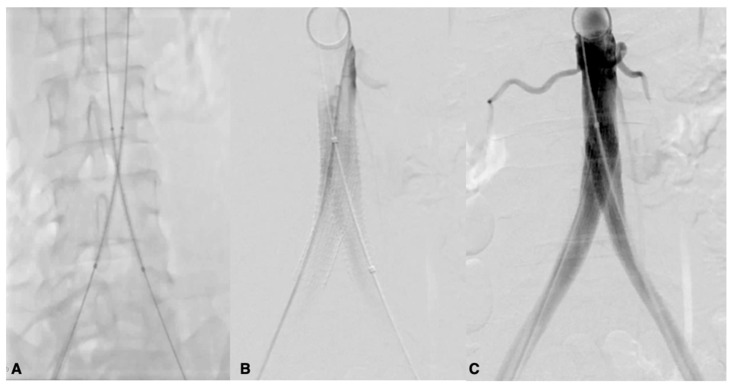
Kissing stent of aortic-iliac biforcation. Placement of the stents (**A**), deployment (**B**), and angiographic check (**C**).

**Figure 7 biomedicines-13-02771-f007:**
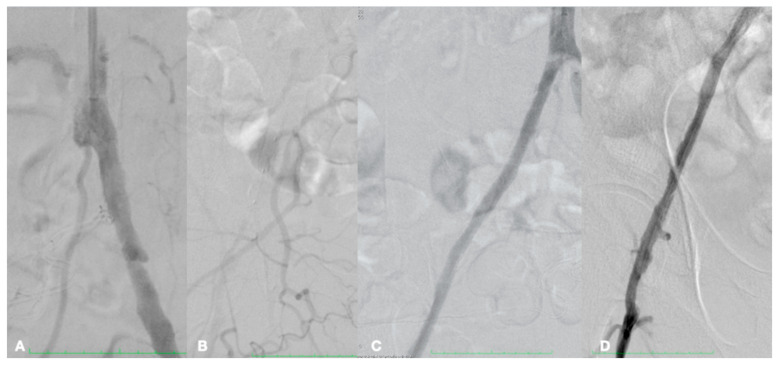
Right iliac axis occluded (**A**,**B**), revascularized through intravascular lithotripsy and drug-eluting balloon (**C**,**D**).

**Figure 8 biomedicines-13-02771-f008:**
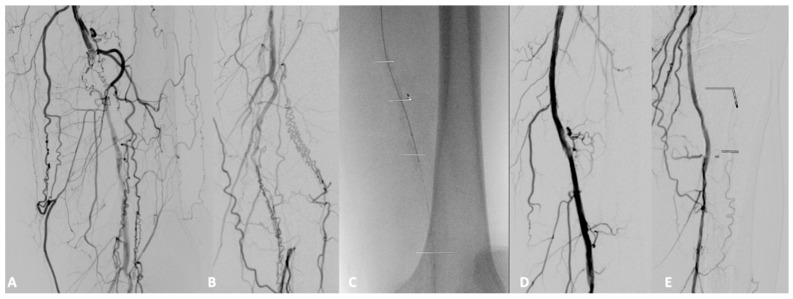
Femoropopliteal axis with occlusion of the distal superficial femoral artery (**A**) and popliteal artery (**B**), revascularized using a debulking system (**C**) and drug-eluting balloon with final check (**D**,**E**).

**Figure 9 biomedicines-13-02771-f009:**
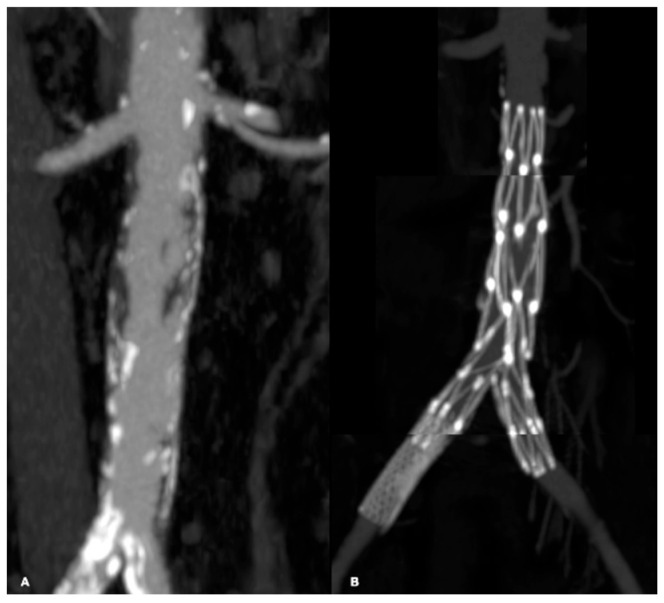
Aortoiliac disease (**A**) treated with implantation of an AFX device and right iliac stent (**B**).

**Figure 10 biomedicines-13-02771-f010:**
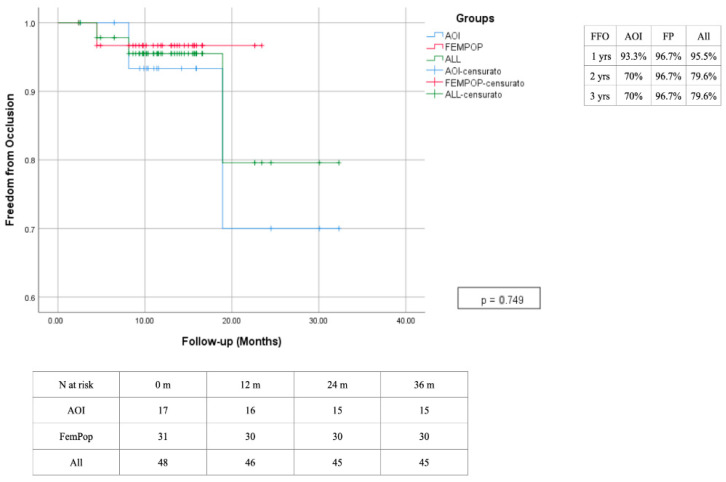
Time freedom reintervention estimated 3-year Kaplan–Meier curves.

**Figure 11 biomedicines-13-02771-f011:**
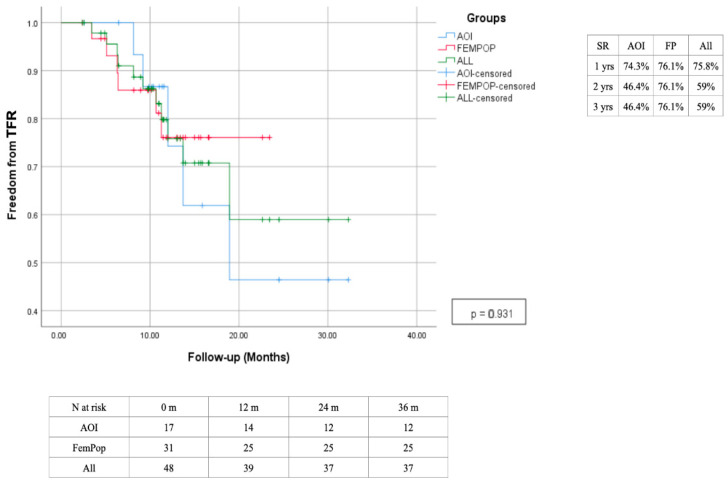
Time freedom from occlusion estimated 3-year Kaplan–Meier curves.

**Figure 12 biomedicines-13-02771-f012:**
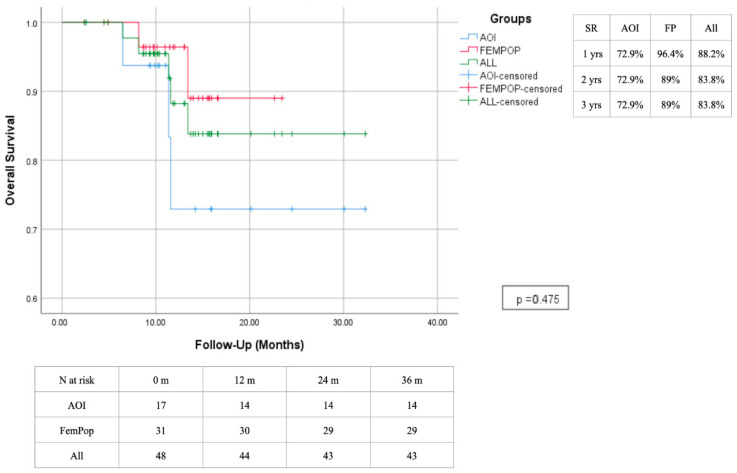
Survival estimated 3-year Kaplan–Meier curves.

**Figure 13 biomedicines-13-02771-f013:**
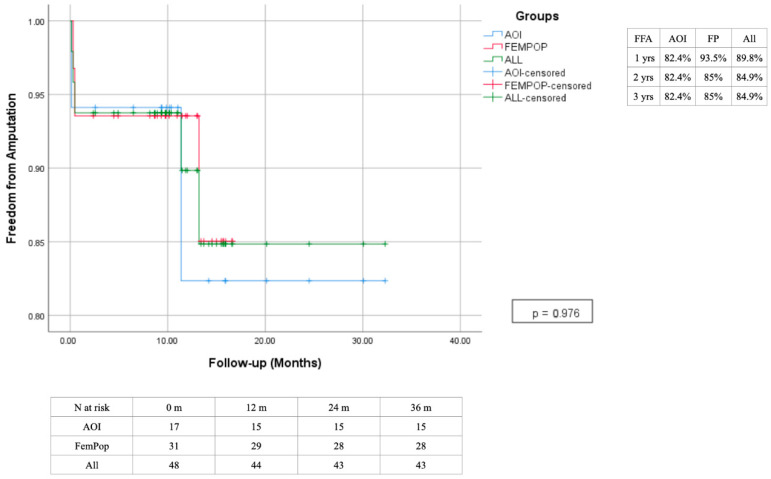
Time freedom from Amputation estimated 3-year Kaplan–Meier curves.

**Figure 14 biomedicines-13-02771-f014:**
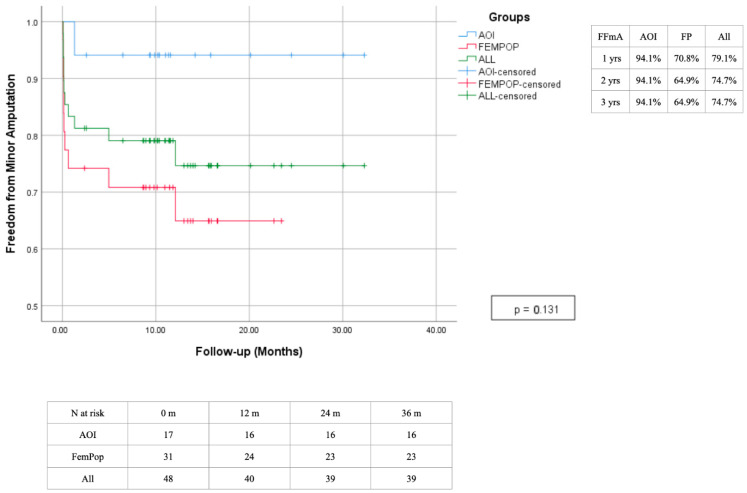
Time freedom from minor amputation estimated 3-year Kaplan–Meier curves.

**Figure 15 biomedicines-13-02771-f015:**
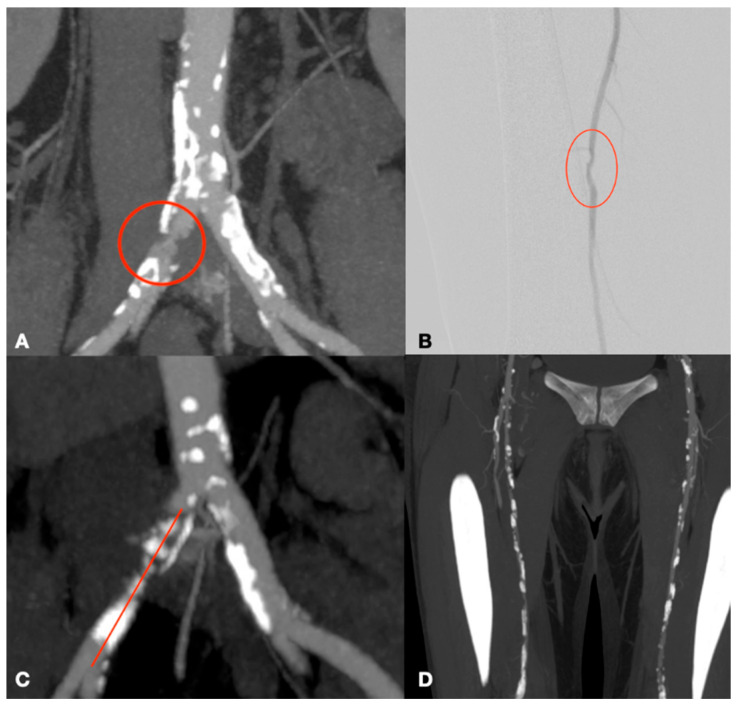
Examples of TASC A and B lesions: TASC A (**A**) in the aorto-iliac district with a stenosis of the common iliac artery less than 3 cm; TASC A (**B**) in the superficial femoral artery with a stenosis less than 5 cm; TASC B (**C**) involving an occlusion of the common iliac artery greater than 3 cm; and TASC B (**D**) affecting the superficial femoral artery with multiple stenoses and segmental occlusions, each less than 5 cm in length.

**Table 1 biomedicines-13-02771-t001:** Nonanatomic patient variables and risk factors.

Risk Factor	Total	Aorto-Iliac	Femoropopliteal	P
N	**48**	**17**	**31**	
Male	30 (62.5%)	11 (64.7%)	19 (61.3%)	0.815
Female	18 (37.5%)	6 (35.3%)	12 (38.7%)	
Mean Age	67.5 ± 6.3	64.3 ± 5.1	69.2 ± 7.4	**0.018**
Cerebrovascular disease	7 (14.6%)	0 (0%)	7 (22.6%)	**0.034**
Hypertension	37 (77.1%)	12 (70.6%)	25 (80.6%)	0.428
Diabetes	32 (66.7%)	8 (47.1%)	24 (77.4%)	**0.033**
Ischemic heart disease	16 (33.3%)	6 (35.3%)	10 (32.3%)	0.831
PCI/CABG	11 (22.9%)	4 (23.5%)	7 (22.6%)	0.940
Oncologic Disease	5 (10.4%)	4 (23.5%)	1 (3.2%)	**0.028**
Dialysis	1 (2.1%)	0 (0%)	1 (3.2%)	0.454
Tobacco Use	27 (56.3%)	9 (52.9%)	18 (58.1%)	0.732
Renal Failure	30 (62.5%)	12 (70.6%)	18 (58.1%)	0.391

**Table 2 biomedicines-13-02771-t002:** Therapy before procedure.

Therapy	Total	Aorto-Iliac	Femoropopliteal	P
N	**48**	**17**	**31**	
Statin	31 (64.6%)	11 (64.7%)	20 (64.5%)	0.990
ASA	15 (31.3%)	3 (17.6%)	12 (38.7%)	0.132
Clopidogrel	8 (16.7%)	4 (23.5%)	4 (12.9%)	0.345
DAPT	22 (45.8%)	8 (47.1%)	14 (45.2%)	0.900
TAO	6 (12.5%)	6 (35.3%)	0 (0%)	**0.001**
NOAC	7 (14.5%)	0 (0%)	7 (22.6%)	**0.034**

**Table 3 biomedicines-13-02771-t003:** Distribution of patients according to the Rutherford, TASC, and ASA classifications.

	Total	Aorto-Iliac	Femoropopliteal	P
N	**48**	**17**	**31**	
**Rutherford**				
Rutherford 3, n (%)	7 (14.6%)	5 (29.4%)	2 (6.5%)	**0.031**
Rutherford 4, n (%)	10 (20.8%)	6 (35.3%)	4 (12.9%)	0.068
Rutherford 5, n (%)	26 (54.2%)	5 (29.4%)	21 (67.7%)	**0.011**
Rutherford 6, n (%)	5 (12.5%)	1 (5.9%)	4 (12.9%)	0.446
**TASC**	6 (12.5%)	6 (35.3%)	0 (0%)	**0.001**
TASC C, n (%)	23 (47.9%)	8 (47.1%)	15 (48.4%)	0.930
TASC D, n (%)	25 (52.1%)	9 (52.9%)	16 (51.6%)	0.930
**ASA**				
ASA III, n (%)	47 (97.9%)	17 (100%)	30 (96.8%)	0.454
ASA IV, n (%)	1 (2.1%)	0 (0%)	1 (3.2%)	0.454

**Table 4 biomedicines-13-02771-t004:** Distribution of vascular lesions.

	Total	Aorto-Iliac	Femoropopliteal	P
N	**210**	**47**	**163**	
**Aorta, n %**	6 (2.9%)	6 (12.8%)	0 (0%)	
Occlusion, n (%)	4 (1.9%)	4 (8.5%)	0 (0%)	**0.005**
Stenosis, n (%)	2 (1%)	2 (4.3%)	0 (0%)	0.051
**Common Iliac Artery**	23 (10.9%)	23 (48.9%)	0 (0%)	
Occlusion, n (%)	11 (5.2%)	11 (23.4%)	0 (0%)	**0.001**
Stenosis, n (%)	12 (5.7%)	12 (25.5%)	0 (0%)	**0.001**
**External Iliaca Artery**	12 (5.7%)	12 (25.5%)	0 (0%)	
Occlusion, n (%)	5 (2.4%)	5 (10.6%)	0 (0%)	**0.001**
Stenosis, n (%)	7 (3.3%)	7 (14.9%)	0 (0%)	**0.001**
**Internal Iliaca Artery**	6 (2.9%)	6 (12.8%)	0 (0%)	
Occlusion, n (%)	3 (1.4%)	3(6.4%)	0 (0%)	**0.016**
Stenosis, n (%)	3 (1.4%)	3(6.4%)	0 (0%)	0.051
**Common Femoral Artery**	13 (6.2%)	6 (12.8%)	7 (4.3%)	
Occlusion, n (%)	3 (1.4%)	2 (4.3%)	1 (0.6%)	0.242
Stenosis, n (%)	10 (4.8%)	4 (8.5%)	6 (3.7%)	0.733
**Superficial Femoral Artery**	29 (13.8%)	0 (0%)	29 (17.8%)	
Occlusion, n (%)	17 (8.1%)	0 (0%)	17 (10.4%)	**0.001**
Stenosis, n (%)	12 (5.7%)	0 (0%)	12 (7.4%)	**0.003**
**Popliteal Artery**	29 (13.8%)	0 (0%)	29 (17.8%)	
Occlusion, n (%)	13 (6.2%)	0 (0%)	13 (8%)	**0.002**
Stenosis, n (%)	16 (7.6%)	0 (0%)	16 (9.8%)	**0.001**
**Anterior Tibial Artery**	27 (12.9%)	0 (0%)	27 (16.6%)	
Occlusion, n (%)	21 (10%)	0 (0%)	21 (12.9%)	**0.001**
Stenosis, n (%)	6 (2.9%)	0 (0%)	6 (3.7%)	0.052
**Tibio-peroneal trunk**	21 (10%)	0 (0%)	21 (12.9%)	
Occlusion, n (%)	11 (5.2%)	0 (0%)	11 (6.8%)	**0.005**
Stenosis, n (%)	10 (4.8%)	0 (0%)	10 (6.1%)	**0.008**
**Peroneal Artery**	16 (7.6%)	0 (0%)	16 (9.8%)	
Occlusion, n (%)	8 (3.8%)	0 (0%)	8 (4.9%)	**0.022**
Stenosis, n (%)	8 (3.8%)	0 (0%)	8 (4.9%)	**0.022**
**Posterior Tibial Artery**	28 (13.3%)	0 (0%)	28 (17.2%)	
Occlusion, n (%)	26 (12.4%)	0 (0%)	26 (16%)	**0.001**
Stenosis, n (%)	2 (1%)	0 (0%)	2 (1.2%)	0.285

**Table 5 biomedicines-13-02771-t005:** Distribution of Treatments.

	Total	Aorto-Iliac	Femoropopliteal	P
N	**48**	**17**	**31**	
PTA	32 (66.7%)	6 (35.3%)	26 (83.9%)	**0.001**
Stenting	24 (50%)	16 (94.1%)	8 (25.8%)	**0.001**
Bebulking	4 (8.3%)	0 (0%)	4 (12.9%)	0.122
Unibody AFX	2 (4.2%)	2 (11.8%)	0 (0%)	0.051
Intravascular Lithotripsy	6 (12.5%)	1 (5.9%)	5 (16.1%)	0.052
Hybrid procedure	30 (62.5%)	9 (52.9%)	21 (67.7%)	0.311

## Data Availability

Data presented in this study is contained within the article. Further inquiries can be directed to the corresponding author.

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
