# Peer review of "Endovascular Treatment Outcomes for TASC C and D Lesions in Chronic Peripheral Arterial Disease: A Retrospective Study and Literature Review"

_biomedicines, 2025, doi:10.3390/biomedicines13112771_

Round 1

Reviewer 1 Report (New Reviewer)

Comments and Suggestions for Authors

The authors reported their interventional results of peripheral vascular disease. They included 48 patients with TASC C-D ilio-femoral lesion. It was single center resitry. The overall presentation of the article is good and results are comparable to medical literature. Small sample size is the major limitation for this study. They performed 12 months follows and their follow up results are also comparable with medical literature. Additon of  the more cases would enhance the scientific value of the the paper.

Author Response

Thank you for the review and the comment. We will try to report an experience with a larger number of cases in future studies.

Reviewer 2 Report (New Reviewer)

Comments and Suggestions for Authors

this is large, comprehensively described, single center series of endovascular interventions for tasc c and d lesions. 

  • can you describe whether covered or bare metal stents were used, particularly in the iliac vessels?
  • the indication for the procedures (claudication vs CLTI) would be helpful
  • multivariate analysis to shed light on complications or outcomes would be an additional point from this data. 
Comments on the Quality of English Language

Reasonable quality - could use light editing for readability. 

Author Response

Comment1: Covered stents were primarily used to treat lesions at the aortic bifurcation and common iliac arteries, with balloon-expandable stents being the predominant choice in these locations. For the external iliac arteries, bare metal stents, mostly self-expanding, were favored. In the femoral region, self-expanding bare metal stents were commonly deployed, while covered self-expanding stents were reserved for cases involving the popliteal artery. The use of covered stents in treating lesions at the aortic bifurcation and common iliac arteries is well supported by clinical evidence, as these devices provide a barrier against plaque prolapse and reduce the risk of embolization, which is particularly important in larger-caliber vessels with complex anatomy. Balloon-expandable covered stents are often preferred in these locations due to their precise deployment and radial strength, which ensures adequate vessel scaffolding. In contrast, bare metal stents, predominantly self-expanding types, are commonly employed in the external iliac arteries due to their flexibility and adaptability to vessel movement, reducing fracture risk and improving long-term patency. Self-expanding bare metal stents are also favored in the femoral region, where vessel mobility and bending are significant factors.

Comment 2: I have clarified this concept better in the text. The patients included in the study were affected by peripheral arterial disease (PAD), experiencing severely limited walking distances of just a few meters or presenting with critical limb ischemia.

COmment 3: We will consider a multivariate analysis for another article. The structure of the current paper does not allow for this analysis, which requires cross-referencing raw data for each individual patient.          

Reviewer 3 Report (New Reviewer)

Comments and Suggestions for Authors

Great job and results!

Introduction

No figures in this section

Materials and Methods

Explain more about the endovascular procedures, such as why you used an aortic endoprosthesis

Small sample of patients

Enrich it with the treatment methods mentioned and add figures

Report the follow-up exams and the technique of reassessment

Discussion

Extensive discussion with repetitions. Focus on the main points

Conclusion

Lack of clarity in the conclusion

Repetition

Comments on the Quality of English Language

Edit the grammar

Author Response

I have followed all the instructions, thank you.

Reviewer 4 Report (New Reviewer)

Comments and Suggestions for Authors

Thank you for doing this; your article was a pleasure to read. I hope my comments will help you further strengthen it.

1-Previous vascular surgery for atherosclerotic disease of the lower limbs was noted in 25 patients (52.1%), with 18 (37.5%) undergoing endovascular procedures and 7 (14.6%) undergoing surgical interventions. The interventions previously performed on these patients differed from those for which the subjects were included in the study. Were any procedures performed on these patients that would affect your statistical results? If you believe this is the case, this should be noted in the limitations section.

2-(results section page 14/19)14 cases required amputation. What was the reason for these patients' amputation? Was it failure during the procedure or a subsequent condition? Which patients were more likely to undergo amputation? Were balloons used or stents placed? This area requires further clarification.

3-The primary patency rate was stated as 75.8% at 12 months. Fourteen patients underwent amputation, and a total of 48 patients were included in the study. What was the criterion used to calculate the success rate? If vascular patency was the success criterion, what method was used to determine this? Did the patients undergoing amputation have vascular patency?

Author Response

Comment 1: The cases of previously treated patients did not affect the statistical results related to the type of intervention, but only influenced the choice of the revascularization method. Patients whose revascularization outcome was compromised by a prior procedure were excluded.

comment 2: I have clarified this concept further in the text. Minor amputations observed in this study resulted from successful revascularization that averted the need for more extensive tissue removal. In cases requiring major amputation, patients presented with severely compromised baseline conditions that either prevented optimal revascularization or led to a gradual loss of the procedure’s benefits over time.

comment 3: The success rate of revascularization was determined by ultrasound data and clinical improvement. Patients who underwent minor amputation had an improved vascular status with preserved patency, while those who underwent major amputation represented either a failure or a consequence of a slow and progressive deterioration following initial revascularization success.

Reviewer 5 Report (New Reviewer)

Comments and Suggestions for Authors

This version is ready for publication. 

Author Response

thanks

Round 2

Reviewer 4 Report (New Reviewer)

Comments and Suggestions for Authors

Thank you for doing such a research, The author has made the necessary corrections and I think the publication quality of the study has increased.

This manuscript is a resubmission of an earlier submission. The following is a list of the peer review reports and author responses from that submission.

Round 1

Reviewer 1 Report

Comments and Suggestions for Authors

Dear authors,

Congratulations on your work. It was a pleasure to read your paper. I hope my comments help you improve your paper:

1. Abstract

1.a. Line 12. I have never heard of chronic peripheral obliterating arteriopathy, but I've heard PAD a million times. I advice you use PAD instead of the other term.

1.b. Line 17. Change "investigates" to "describes".

1.c. Lines 26-27. Change "can be a viable alternative" to "may be a viable alternative".

1.d. Line 28. Remove "reducing the need for more invasive surgical procedures". You don't have data to sustain this claim.

1.e. Remove "emphazising the necessity for updated clinical guidelines". You don't have data to support changing guidelines. Your study is a descriptive study, not a comparative effectiveness study.

2. Introduction:

2.a. Your introduction is too long. It almost seems like a review of PAD. I recommend you change it to two paragraphs: the first paragraph should BRIEFLY introduce PAD and the treatment strategies to manage it (i.e., open vs. endo). The second paragraph should introduce the TASC classification as a tool that guides management of PAD, and state that TASC A and B lesions are usually managed with endovascular therapy while C and D lesions are usually managed with open surgery, but your team manages some (or all?) TASC C and D lesions with endovascular therapy and you are presenting your data on this experience. The text should fit in a single page. Your first three paragraphs should be condensed into a single paragraph. There are some other details regarding grammar but change the structure first and I can give you more feedback on grammar once you have a shorter version of the introduction.

3. Materials and Methods

3.a. Line 86. You are not comparing endovascular therapy to open therapy for TASC C and D lesions, you are describing your experience using endovascular therapy for these lesions. I would argue that saying you are "evaluating the effectiveness" implies comparison to the standard of care (i.e., open surgery). Please change this sentence so it reads "This is a non-randomized retrospective study aimed at describing our experience using endovascular treatment for TASC C and D lesions, for which surgical intervention is typically indicated."

3.b. Line 120-121. Mean and SD are used for continuous parametric data, medians and ranges are used for continuous non-parametric data. Absolute values and percentages are used for categorical data. Please correct your sentence.

3.c. Line 124. Please eliminate the sentence "Statistical significance was considered at <0.05." This is an arbitrary cut-off. Presenting the point estimates, measures of spread, and p-values should suffice.

3.d. Line 126. SPSS Inc. was acquired by IBM in 2010. Please change your text to "SPSS 16.0 (IBM, New York, NY, USA)."

4. Results

4.a. Line 128. No need to say 18 were female and 30 male. Its redundant. Just say "The study comprised 48 patients, of whom 18 were female (37.5%)."  

4.b. Table 1. Remove "anni" add the standard deviation for mean age.

5. Discussion

5.a. Line 223. Change "Our study aimed to evaluate..."  to "Our study aimed to describe..."

6. Conclusions

6.a. Your study doesn't present a comparison of endovascular therapy to open surgery, and your sample size is small and done at a single center. Therefore you cannot say that endovascular therapy "is" a viable option for TASC C/D lesions. You can say endovascular therapy "may be" a viable option since you had good outcomes in the experience you are presenting, and that is worth exploring this in future studies comparing endo to open surgery for TASC C/D lesions. Change the wording of your conclusion to reflect this.

Author Response

1. Abstract

1.a. Line 12. I have never heard of chronic peripheral obliterating arteriopathy, but I've heard PAD a million times. I advice you use PAD instead of the other term.

Response: I corrected this point.

1.b. Line 17. Change "investigates" to “describes".

Response: I corrected this point.

1.c. Lines 26-27. Change "can be a viable alternative" to "may be a viable alternative”.

Response: I corrected this point.

1.d. Line 28. Remove "reducing the need for more invasive surgical procedures". You don't have data to sustain this claim.

Response: I corrected this point.

1.e. Remove "emphazising the necessity for updated clinical guidelines". You don't have data to support changing guidelines. Your study is a descriptive study, not a comparative effectiveness study.

Response: I corrected this point.

2. Introduction:

2.a. Your introduction is too long. It almost seems like a review of PAD. I recommend you change it to two paragraphs: the first paragraph should BRIEFLY introduce PAD and the treatment strategies to manage it (i.e., open vs. endo). The second paragraph should introduce the TASC classification as a tool that guides management of PAD, and state that TASC A and B lesions are usually managed with endovascular therapy while C and D lesions are usually managed with open surgery, but your team manages some (or all?) TASC C and D lesions with endovascular therapy and you are presenting your data on this experience. The text should fit in a single page. Your first three paragraphs should be condensed into a single paragraph. There are some other details regarding grammar but change the structure first and I can give you more feedback on grammar once you have a shorter version of the introduction.

Response: I have corrected the introduction; I hope I have met your instructions.

3. Materials and Methods

3.a. Line 86. You are not comparing endovascular therapy to open therapy for TASC C and D lesions, you are describing your experience using endovascular therapy for these lesions. I would argue that saying you are "evaluating the effectiveness" implies comparison to the standard of care (i.e., open surgery). Please change this sentence so it reads "This is a non-randomized retrospective study aimed at describing our experience using endovascular treatment for TASC C and D lesions, for which surgical intervention is typically indicated.”

Response: I corrected this point.

3.b. Line 120-121. Mean and SD are used for continuous parametric data, medians and ranges are used for continuous non-parametric data. Absolute values and percentages are used for categorical data. Please correct your sentence.

Response: I corrected this point.

3.c. Line 124. Please eliminate the sentence "Statistical significance was considered at <0.05." This is an arbitrary cut-off. Presenting the point estimates, measures of spread, and p-values should suffice.

Response: I corrected this point.

3.d. Line 126. SPSS Inc. was acquired by IBM in 2010. Please change your text to "SPSS 16.0 (IBM, New York, NY, USA).”

Response: I corrected this point.

4. Results

4.a. Line 128. No need to say 18 were female and 30 male. Its redundant. Just say "The study comprised 48 patients, of whom 18 were female (37.5%)."  

Response: I corrected this point.

4.b. Table 1. Remove "anni" add the standard deviation for mean age.

Response: I corrected this point.

5. Discussion

5.a. Line 223. Change "Our study aimed to evaluate..."  to "Our study aimed to describe…"

Response: I corrected this point.

6. Conclusions

6.a. Your study doesn't present a comparison of endovascular therapy to open surgery, and your sample size is small and done at a single center. Therefore you cannot say that endovascular therapy "is" a viable option for TASC C/D lesions. You can say endovascular therapy "may be" a viable option since you had good outcomes in the experience you are presenting, and that is worth exploring this in future studies comparing endo to open surgery for TASC C/D lesions. Change the wording of your conclusion to reflect this.

Response: I corrected this point.

Thank you for carefully reading my work and for your valuable corrections

Reviewer 2 Report

Comments and Suggestions for Authors

Reviewer Comments

  1. Page 1, lines 24-25: What was the average length of time from intervention to amputation for the 33.3% of patients that required amputation? This is a significant amount post-intervention.
  2. Page 3, lines 86-88: For future research it would be good to have a comparison group of patients who underwent open surgical interventions. But given the common comorbidities of vascular patients, it is understandable that this could be challenging to achieve.
  3. Page 5, line 139: Can the authors please clarify what “Thromboembolic prevention therapy (TAO)” is for the readers?
  4. Page 5, lines 144-146: For the patients who underwent prior endovascular procedures, 18 (37.5%), how many of these were the same area intervened on in the study/re-ops?
  5. Page 7, line 157 & Page 8, line 168: It seems very low that only 50% of patient’s received stents during intervention. For TASC C & D lesions in the aorto-iliac and SFA/popliteal areas this is high chance of re-stenosis with angioplasty alone. Particularly, for calcified iliac disease, stenting is routinely used.  Can the authors comment on their centers practice and reasoning for less use of endovascular stenting
  6. Page 8, line 180: What was the case that required tibial artery access? Was it a fem-pop intervention?
  7. Page 8, lines 195-196: Can the authors clarify what the “inferior renal artery” complication was in acute complications.
  8. Page 9, lines 202-205: There were high re-internvetion rates for the aortoiliac group, but what percentage of this group underwent iliac stenting? If this percentage is low it could be a potential explanation .
  9. Page 10, lines 214-215: Can the authors give a definition of minor amputation (toe amp and TMA?) and major amputation (BKA vs AKA?) for the reader?

Author Response

  1. Page 1, lines 24-25: What was the average length of time from intervention to amputation for the 33.3% of patients that required amputation? This is a significant amount post-intervention.

Response: I have added a clarification to the discussion. First of all, I identified an error in the previous calculation: the amputation rate is actually slightly lower at 29.2% rather than 33.3%. Most of the amputations (12 out of 14) occurred within the first year. Considering that approximately 66.7% of patients had trophic lesions at the time of admission, this figure is acceptable.

  1. Page 3, lines 86-88: For future research it would be good to have a comparison group of patients who underwent open surgical interventions. But given the common comorbidities of vascular patients, it is understandable that this could be challenging to achieve. ok
  2. Page 5, line 139: Can the authors please clarify what “Thromboembolic prevention therapy (TAO)” is for the readers? 

Response: I have added this point to the text.

  1. Page 5, lines 144-146: For the patients who underwent prior endovascular procedures, 18 (37.5%), how many of these were the same area intervened on in the study/re-ops?

Response: I have added this point to the text.

  1. Page 7, line 157 & Page 8, line 168: It seems very low that only 50% of patient’s received stents during intervention. For TASC C & D lesions in the aorto-iliac and SFA/popliteal areas this is high chance of re-stenosis with angioplasty alone. Particularly, for calcified iliac disease, stenting is routinely used.  Can the authors comment on their centers practice and reasoning for less use of endovascular stenting

Response: I have added this point to the text.

  1. Page 8, line 180: What was the case that required tibial artery access? Was it a fem-pop intervention?

Response: We are accustomed to performing a tibial retrograde access for popliteal revascularizations when an antegrade revascularization is not feasible.

  1. Page 8, lines 195-196: Can the authors clarify what the “inferior renal artery” complication was in acute complications.

Response: It was a translation error; I have corrected the text.

  1. Page 9, lines 202-205: There were high re-internvetion rates for the aortoiliac group, but what percentage of this group underwent iliac stenting? If this percentage is low it could be a potential explanation 

Response: I have added this point to the text, but the problem is a greater number of intrastent stenoses.

  1. Page 10, lines 214-215: Can the authors give a definition of minor amputation (toe amp and TMA?) and major amputation (BKA vs AKA?) for the reader?

Response: I have added this point to the text.

Reviewer 3 Report

Comments and Suggestions for Authors

The total number of procedures is relatively small considering you are including iliac and femoral popliteal lesions.  You also stated that a significant number of patients had undergone prior endovascular and open procedures. Were the procedures that you performed in the same anatomic area and why do you believe that your procedures were successful whereas prior procedures failed.  You utilized many methods of intervention i.e DCB, plain PTA, stents, atherectomy.  It is difficult to determine which method had the best result. It also seems that your amputation rate was slightly higher than expected since most procedures were not performed for limb threatening ischemia.  Also you state that there was a high incidence of renal failure.

Author Response

The total number of procedures is relatively small considering you are including iliac and femoral popliteal lesions.  You also stated that a significant number of patients had undergone prior endovascular and open procedures. Were the procedures that you performed in the same anatomic area and why do you believe that your procedures were successful whereas prior procedures failed.  You utilized many methods of intervention i.e DCB, plain PTA, stents, atherectomy.  It is difficult to determine which method had the best result. It also seems that your amputation rate was slightly higher than expected since most procedures were not performed for limb threatening ischemia.  Also you state that there was a high incidence of renal failure.

Response: I made some corrections to the text. The article discusses a retrospective study, along with its inherent limitations. Patients who had already undergone surgery were included, as the new interventions targeted different areas. The number of amputations was lower than the rates reported in the literature for critical ischemia; in our study, they accounted for 87.5% of the patients. The data on renal failure refer to the preoperative period. We documented only one case of acute renal failure.

Round 2

Reviewer 3 Report

Comments and Suggestions for Authors

Despite the revisions I think that the paper has too small a sample size and rally does not add to the literature.

Author Response

In this article, we share our experience and emphasize the importance of enhancing endovascular techniques, driven by advancements in instrumentation.